



# Ecohydrological particle model based on representative domains

Conrad Jackisch[1] and Erwin Zehe[1]

[1]Karlsruhe Institute of Technology KIT, Institute of Water Resources and River Basin Management, Chair of Hydrology, Kaiserstr. 12, 76131 Karlsruhe, Germany.

*Correspondence to:* Conrad Jackisch, Karlsruhe Institute of Technology KIT, Institute of Water Resources and River Basin Management, Chair of Hydrology, Kaiserstr. 12, 76131 Karlsruhe, Germany. (jackisch@kit.edu)

**Abstract.** Non-uniform infiltration and subsurface flow in structured soils is observed in most natural settings. It arises from imperfect lateral mixing of fast advective flow in structures and diffusive flow in the soil matrix and remains one of the most challenging topics with respect to match observation and modelling of water and solutes at the plot scale.

This study extends the fundamental introduction of a space-domain random walk of water particles as alternative approach

to the Richards equation for diffusive flow (Zehe and Jackisch, 2016) to a stochastic-physical model framework simulating soil water flow in a representative, structured soil domain. The central objective of the proposed model is the simulation of non-uniform flow fingerprints in different ecohydrological settings and antecedent states by making maximum use of field observables for parameterisation. Avoiding non-observable parameters for macropore-matrix exchange, an energy-balance approach to govern film flow in representative flow paths is employed. We present the echoRD model (ecohydrological particle

model based on representative domains) and a series of application test cases.

The model proves as a powerful alternative to existing dual-domain models, driven on experimental data and with self-controlled, dynamic macropore-matrix exchange from the topologically semi-explicitly defined structures.

## 1 Introduction

Non-uniform subsurface flow is omnipresent in hydrology (Uhlenbrook, 2006) and is today accepted as being the rule rather

than the exception (Flury et al., 1994; Nimmo, 2011). Originally, preferential flow describes water transport in non-capillary soil structures which is much faster than would be expected from classical theory of flow and transport in porous media (e.g. Bear, 1975). A considerable number of studies and model approaches have since been proposed to address the issue – as explained in several reviews (especially Beven and Germann, 1982; Šimůnek et al., 2003; Jarvis, 2007; Weiler and McDonnell, 2007; Köhne et al., 2009b; Beven and Germann, 2013).

Macropore settings may be very specific with respect to their topology, their temporal dynamics and their interface characteristics in their ecohydrological context: Earthworm burrow configurations (Blouin et al., 2013), their spatio-temporal dynamics (Palm et al., 2012; van Schaik et al., 2014) and burrow coatings (Jarvis, 2007; Rogasik et al., 2014) affect infiltration and water redistribution. Also other structure-creating animals like rodents and moles can have an impact (Botschek et al., 2002). Plant roots affect water redistribution and soil water withdraw dynamically (Nadezhdina et al., 2010). Connected flow paths





(Wienhöfer, 2014) and periglacial cover beds (Heller, 2012) may change the hydrological regime completely.

All of these influences are rather complex and specific in detail. In addition, they challenge the model concepts since the advective processes take place in explicit structures with respective connectivity and spatial covariance and under far from well-mixed conditions. They extend across several scales in space and time.

Non-uniform flow arises from imperfect lateral mixing between a fast advective fraction of water and solutes (traveling mainly driven by gravity in large pores and soil structures) and a slow diffusive fraction (governed by capillary forces in the soil matrix) (Blöschl, 2005; Neuweiler and Vogel, 2007). Advective flow in structures is governed by initial supply (Weiler, 2005) and interaction with the soil matrix (Nimmo, 2016; Germann and Karlen, 2016). Thus, interaction comprises the exchange of mass and dissipation of flow kinetic energy. The proposed approaches to deal with this deviation from local equilibrium state range from a) the early concept of stochastic convection i.e. no mixing at all (Jury and Roth, 1990), or b) with mixing as multiple interacting pathways (Davies et al., 2013), over c) the scale way idea to convey structural fingerprints in flow and transport across scales (Vogel and Roth, 2003), d) dual porosity/permeability approaches relying on overlapping and exchanging continua (Gerke, 2006), to e) spatially explicit or representative definition of macropores as vertically and laterally connected flow paths based on elevated conductivity (Vogel et al., 2006; Sander and Gerke, 2009; Klaus and Zehe, 2011). In particular the last approach corroborates the crucial importance of reliable field data or estimates characterising the distribution of the macropores at the surface and over depth for successful predictions (Loritz et al., 2017). In addition, their potential connection to lateral preferential flow paths and the catchment drainage network is of fundamental interest (Jackisch et al., 2017).

Kleidon et al. (2013) and Zehe et al. (2013) have focused on the role of preferential flow from an energy or momentum perspective. While preferential flow hinders lateral mixing, it facilitates vertical mass transfer against differences in geo-potential or large gradients in matrix potential, which establish during dry spells in cohesive soils and lead to a faster depletion of the gradients (Westhoff et al., 2014). This implies a faster reduction (dissipation and export) of free energy of soil water during rainfall driven conditions due to enhanced mixing into the main direction of the flow path (Zehe et al., 2013). Also exchange between both flow domains is associated with dissipation of kinetic energy and thus momentum (Kutilek and Germann, 2009).

Despite the fact that there has been considerable progress in the understanding of preferential flow and non-uniform infiltration, the topic remains one of the most challenging in particular with respect to scale and sub-scale representation of rapid subsurface flow and transport in hydrological models (Beven and Germann, 2013) and with respect to feedbacks between soil ecology and soil hydrology (van Schaik et al., 2014).

We thus propose a stochastic-physical model framework to jointly predict rapid advective water flows in soil structures, diffusive water flows when capillarity controls soil water dynamics, and the interaction between the two. The approach is developed for a representative plot domain with topologically explicit macropores. An overall goal of the model framework is to open up ways to virtual experiments on infiltration patterns and abiotic controls on specific niches for macro- and microbiota





in structured subsurface domains.

The proposed model is a Lagrangian approach treating water itself as particles moving diffusively by means of a space domain random walk and advectively as film flow in representative structures. Lagrangian approaches to solute transport and unsaturated flow in heterogeneous media are well established tools in hydrological modelling (among others Neuweiler et al., 2012; Delay and Bodin, 2001). Most particle tracking applications calculate the water flow as external drift based on a hydrological solver like for the Richards equation (de Rooij et al., 2013) or establish some assumption about the fate of a random walker in the time domain (Dentz et al., 2012). Lagrangian approaches to plot- and hillslope-scale water dynamics itself were to our knowledge only followed by Ewen (1996) (subsystems and moving packets model) and Davies et al. (2011) (multiple interacting pathways model). However, both approaches solve the key problem of advective momentum dissipation by macropore-matrix interaction by means of explicit parameterisation. While Ewen (1996) introduces different types of water movement with a structural property parameter $\lambda$ to govern the probability of a water particle to move, Davies et al. (2011) define an exchange or mixing parameter of the particles' "momentums". Both approaches have proven very suitable for their application. Yet, both parameters remain to be estimated by calibration. This implies strong limitations for predictions in dynamic systems and systems under change.

We have shown in a previous study (Zehe and Jackisch, 2016) that the space domain random walk (1D) allows for a physically consistent representation of capillarity-driven, unsaturated soil water flow in accordance with the Richards equation. Here, we extend the approach to a 2D matrix domain which hosts a number of representative preferential flow structures like earthworm burrows or cracks as vertical 1D elements. The scope of this echoRD model (eco-hydrological particle model based on representative structured domains) is on simulation of plot scale flow and transport through an explicit treatment of macropores. Pore scale processes (e.g. Moebius and Or, 2012; Shahraeeni and Or, 2012; Snehota et al., 2015; Schlüter et al., 2016) are not resolved here.

The main objectives of this study are to a) present the model theory, to b) test the capability of the echoRD model to simulate the fingerprints of plot scale non-uniform infiltration and to c) reveal whether advective and diffusive flow and the interactions among those may be represented in one consistent formulation. As the model shall allow for virtual experiments we base its parameterisation as much as possible on field observables or explicitly testable hypotheses. More specifically, we derive and test an energy-based approach to control the exchange between the macropore domain and the surrounding matrix in a self-limiting manner.

The software developed and data used in this study are available under GNU General Public License (GPLv3) and Creative Commons License (CC BY-NC-SA 4.0) respectively through a git repository: https://github.com/cojacoo/echoRD_model. In particular, the echoRD model, including a preprocessor, application tests and basic documentation can be accessed there.



## 2 Specific motivation

### 2.1 General particle concept and 1D implementation

Particle tracking is usually employed for simulating advective dispersive transport of solutes, but not for the water phase itself (e.g. Delay and Bodin, 2001; Metzler and Klafter, 2004; Berkowitz et al., 2006; Koutsoyiannis, 2010). Thus, most random walk applications rely on a continuous time domain representation as it performs well at minimum computational cost (Delay et al.,
2008; Dentz et al., 2012). This approach is, however, not feasible when the diffusivity itself depends on the particle density as is the case for water particles. We thus employ a non-linear random walk of water particles in the space domain.

    In Zehe and Jackisch (2016) we described this 1D model with water particles of constant mass traveling according to the Itô form of the Fokker Planck equation. The model concept builds on established soil physics by estimating the drift velocity and the diffusion term based on the soil water retention characteristics. Reduced mobility of water with decreasing
pore size is accounted for using a suitable binning of the water diffusivity curve to scale the random work of different particles. Furthermore, we proposed a straightforward implementation of rapid non-equilibrium infiltration there. Event water is treated as different type of particles, which travel initially in the largest pore fraction at maximum velocity and experience a slow diffusive mixing with the pre-event water particles within a characteristic mixing time.

### 2.2 Limitations of the 1D representation

Despite the successful application of the introduced particle model approach, a 1D version essentially lacks information about the lateral component of the non-uniform distribution and resulting macropore-matrix exchange characteristics. In order to unify an essence of the recent model approaches for subsurface flow in discrete structures (e.g. Jury and Roth, 1990; Vogel and Roth, 2003; Gerke, 2006; Vogel et al., 2006; Sander and Gerke, 2009; Nimmo, 2011), the most simple model, which adds a third type of particles to our previous 1D representation, thus would imply three problems:

The first is that macropore flow is much faster than saturated hydraulic conductivity. At the same time it is limited to a very small fraction of the soil column. This motivated the conceptualisation of multiple flow domains. However, the state of a specific flow path is substantially different from the averaged state of a elementary volume. Secondly, the topology of flow paths plays a role in this regard: Macropores enable a quick vertical redistribution of event water. If the network of macropores is rather dense and lateral diffusion is not too slow, the resulting soil water dynamics can be uniformly described by some
elevated, effective hydraulic conductivity. If the structures are sparse and lateral diffusion into the matrix is slower, lateral gradients in soil water potential and non-uniform flow fields establish.

    As such the flow field depends on macropore topology, antecedent soil matrix state, macropore capacity and infiltration supply. In a 1D approach such lateral gradients and their depletion cannot be described other than by some additional conceptual parameter or function and averaged matric potential states. The result would remain bound to a priori defined macropore-matrix
exchange assumptions. Without proper control of the macropore-matrix interaction and thus control of the advective flow field, a fast fraction of particles would simply remain quick and drain from the domain which contradicts the experimental findings.





The third challenge refers to exchange/mixing of rapid event water particles with the pre-event water to establish local thermodynamic equilibrium (LTE) – the well-organised distribution of water particles in the respective smallest fractions of the available pore space as we further explained earlier (Zehe and Jackisch, 2016).

These issues led to the preliminary finding that a lumped 1D version of the particle model could not succeed in reproducing the observed tracer distributions without thorough calibration to one specific antecedent state and one specific realisation of the advective flow field. The requirement of non-observable and non-static mixing parameters between the domains makes an application to predict behaviour under change challenging. Thus it does not convince much if we desire to develop the model as virtual laboratory.

## 3 The echoRD model

### 3.1 The representative macropore-matrix domain

In order to overcome the 1D limitations without requirement for a pore-scale determination of the macropore system or non-observable parameters, we define a representative macropore-matrix domain with explicit topology (Fig. 1). Soil matrix is projected as 2D domain with a periodic lateral boundary. Macropores are represented as vertical 1D elements linked to the matrix. As there is usually no information about the spatial clustering of macropores, they are placed at resampled distances according to an observed density distribution. Given the periodic lateral boundary of the matrix domain, it is not the macropore positions but their relative distances that matter. The minimum density of the macropores at a given depth determines the lateral extent of the domain. One may also chose to take a multiple of the least representative as setup for instance to describe interactions with less densely occurring structures such as subsurface pipes.

The 2D soil matrix possesses a grid for particle density calculation. The 1D macropore domains have an internal grid for film flow calculations, where the lag distance is calculated as projection of one water particle to the mean macropore diameter. In addition, the 1D macropore domains have an interface area with the 2D soil matrix domain. In this area particles are considered for exchange between the domains.

### 3.2 Diffusion in the soil matrix based on a 2D random walk

Diffusive soil water flow is simulated as non-linear, space domain random walk as presented in our previous study (Zehe and Jackisch, 2016). We describe the trajectory of a singe particle of water in a time step $\Delta t$ as Itô form of the Fokker Planck equation based on the formal equivalence of the Richards equation and the advection dispersion equation consisting of a trivial drift term $u(\theta_{z,x,t}) = \frac{k(\theta_{z,x,t})}{\theta_{z,x,t}}$ characterising downward water fluxes driven by gravity and a diffusive term representing water movements driven by the matric head gradient and controlled by the diffusivity $D(\theta_{z,x,t})$ of soil water or particles respectively.



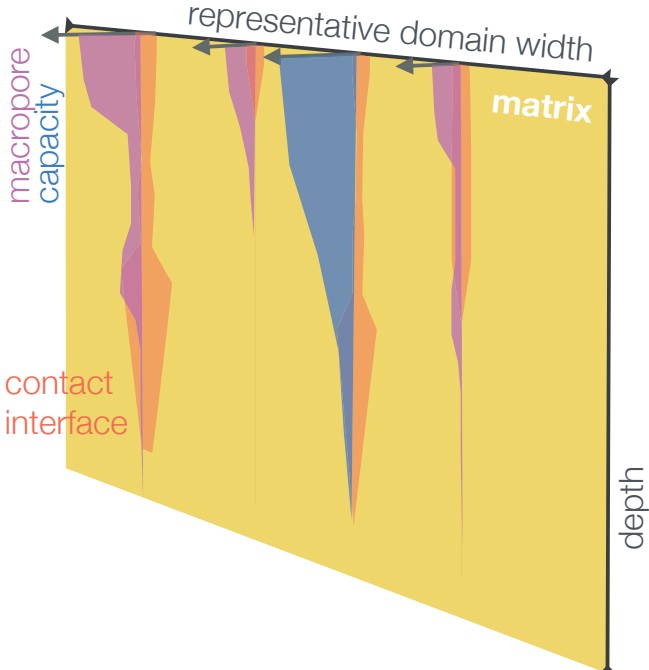

**Figure 1.** Representative macropore-matrix domain. A 2D soil matrix with periodic lateral boundary hosts several 1D macropores with their respective capacities, interfaces and lateral distributions.

With this we can establish the Itô solution for the trajectory of one particle:

$$z_{t+\Delta t} = z_t + \left[ u(\theta_{z,x,t}) + \frac{\partial D(\theta_{z,x,t})}{\partial z_t} \right] \Delta t + \xi_z \sqrt{6D(\theta_{z,x,t})\Delta t}$$

$$x_{t+\Delta t} = x_t + \frac{\partial D(\theta_{z,x,t})}{\partial x_t} \Delta t + \xi_x \sqrt{6D(\theta_{z,x,t})\Delta t} \tag{1}$$

with $z$ vertical position (m), $x$ lateral position (m), and $\xi$ a uniform random number [-1,1]. Notice, that unlike the diffusion/advection of a solute this does not require referencing to the wetted pore space since our reference system is the total pore volume.

In this form diffusivity $D(\theta_{z,x,t})$ is dependent on the soil moisture $\theta$ at the location $(z,x)$ of a particle for a certain time step $(t)$. Although we need to assume point-like particles to apply the Itô solution in equation 1, each particle is referenced to a theoretical spatial extend and mass to derive $\theta$ from the density of particles. However, any kind of direct particle interaction is neglected at this stage. $\theta$ is calculated by counting all particles in the calculation grid of the 2D soil matrix.

Alternatively to the $\theta$-based form which assumes LTE at any time, we can assign each particle to a discrete bin as surrogate of its position in the pore space (Fig. 2). With this it obtains its reference to the water retention curve as explained in Zehe and Jackisch (2016). Then $u$ and $D$ in equation 1 are dependent on the particle's bin. By doing so the re-assignment of bins to the moving particles becomes crucial:

In the advanced model version the bins of all particles in each calculation grid cell are frequently updated by determining the





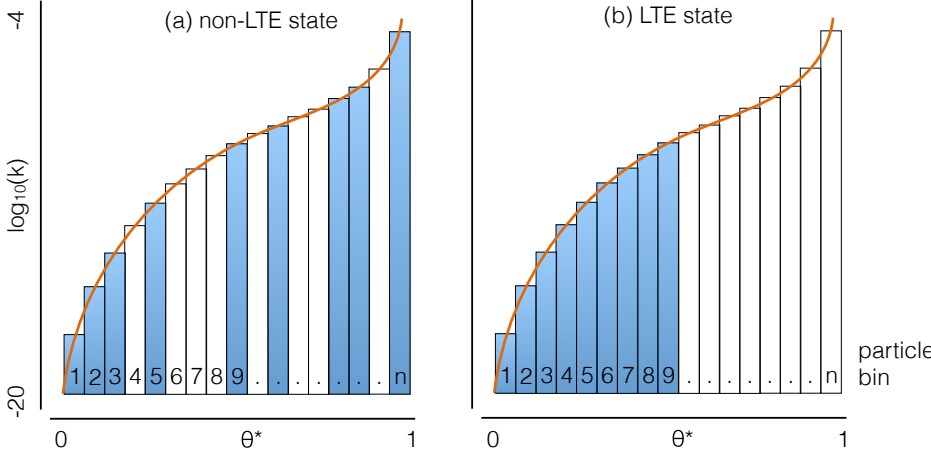

**Figure 2.** Example for delineation of the pore space into bins of equal volumes or particles in our case. If the bins are organised ascendingly, this refers to the LTE (local thermodynamic equilibrium) state of the pore space (b). However, at the same overall soil moisture the particle configuration could also divert from LTE (a).

deviation from LTE state (all bins are sorted from 0 to $n$ with $n$ as number of particles at the current relative saturation state). The relaxation time $t_{mix}$ to LTE is hypothesised as diffusion time:

$$t_{mix} = \frac{L_{x,z}^2}{D_{\text{mix}}} \tag{2}$$

with $L_{x,z}$ as maximal diffusion length given by $L_{x,z} = k_s(x,z) \cdot \Delta t$ and $D_{mix}$ as $D$ at the 0.7 percentile of the free bins (the percentile can controlled by a model parameter). With this, $t_{mix}$ is the time after which LTE is assumed to be recovered from an initial population of the largest pores. The bins of all particles in a grid cell are updated to a lower deviation from LTE after each calculation step by:

$$bin_{t+\Delta t} = bin_t - max\big[0, (bin_t - bin_{\text{LTE}})\big] \frac{\Delta t}{t_{mix}} \tag{3}$$

In addition, a counteracting stochastic process is introduced to handle the effect of high diffusivity but low number of open slots in the pore space near saturation.

$$p_{\text{counteract}} = \frac{n_{\text{empty bins}}}{n_{\text{air capacity bins}}} \tag{4}$$

Here $n$ is the number of respective bins in the pore space. If $p_{\text{counteract}}$ is below 1 it is multiplied with $\xi$ in the random walk (Eq. 1) scaling the diffusive step by the ratio of open slots tending towards zero at saturation.

Numerically, the actual step of a particle is calculated in a predictor-corrector approach projecting the step of one particle, anticipating an updated state to calculate $D$ and $u$ and calculating the geometric mean of the projected and updated $D$ and $u$ after Stratonovich. In oder to balance computational expenses and numerical stability, a stratified subsample (governed by a model parameter) of all particles is handled at once. The used variables are calculated based on van Genuchten parametrisation of the soil matrix properties.





### 3.3 Advection in the 1D macropores as film flow

In addition to the matrix domain the setup contains several 1D elements as macropores (Fig. 1). They are distributed along the lateral axis of the matrix and connect to certain cells over a defined contact interface.

#### 3.3.1 Projected drainage capacity and maximum velocity

The preferential flow network exhibits a large drainage capacity. Zehe (1999) estimated that a single burrow of a *Lumbricus terrestris* ($r = 4.5\,\mathrm{mm}$) may drain the equivalent of $1\,\mathrm{m}^2$ saturated Loess soil matrix. Based on the domain setup, advection is structurally limited by the drainage depth of a macropore and its size.

The second limit is given through the definition of initial maximum flow velocity in the structures. Literature values in Tab. 1 range closely around $7.5 \times 10^{-2}\,\mathrm{m\,s^{-1}}$. Being much larger than the saturated hydraulic conductivity of most soils, these values range several orders of magnitude below the theoretical value for pipe flow in such a pore calculated after Hagen-Poiseuille with a unit gradient. Here we use this difference to estimate frictional losses of the advective momentum as dynamic limitation through interaction with the matrix as further explained in the following sections.

#### 3.3.2 Dynamic film flow

Macropore flow is represented as 1D film flow of particles along the pore wall (Fig. 3). We assume that a particle has a given kinetic energy ($E_{kin}$) which is dissipated by friction at the macropore wall and infiltration into the matrix (Fig. 3A). The maximum advection step $s_{proj}$ of a particle is projected based on its current velocity $v_0$, which is decelerated by the $a_{\mathrm{friction}}$ and $a_{\mathrm{exchange}}$ it experiences along the path. This results in a reduced step length $s_{real}$ (Fig. 3B). On its passage $s_{real}$ a particle may possibly infiltrate into the matrix calculated by an accumulation of an infiltration length (Fig. 3C). We account for variable film thickness depending on the number of particles in each internal grid element. If particles overlap their vertical positions and thus are more than one per position slot, they form a second film layer. Particles at a higher level in a film do not experience drag or friction and travel without retardation until they reach the lowest wetted position within a continuous film stretch (Fig. 3D).

#### 3.3.3 Macropore-Matrix-Interaction

Direct experimental evidence about water dynamics at the macropore-matrix interface hardly exists. Some orientation is given by findings of Hincapié and Germann (2010) and Moebius and Or (2012). Promising techniques like time-lapse X-ray or $\mu$CT tomography just emerge to be applied (Koestel and Larsbo, 2014; Schlüter et al., 2016). Yet there is consensus that macropore-matrix interaction depends on the matric head, the wetting of the macropore wall (Klaus et al., 2013) and is optionally affected by organic coatings which may act hydrophobic (Jarvis, 2007; Rogasik et al., 2014). Moreover, it is dependent upon the flow velocities. Current dual domain approaches treat this key process as either based on a leakage/exchange coefficient and the potential difference between the domains (Gerke, 2006) or by using the geometric mean of the saturated hydraulic and actual hydraulic conductivity and the potential gradient between both domains. The latter depends on an exchange length (Beven

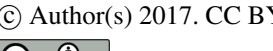



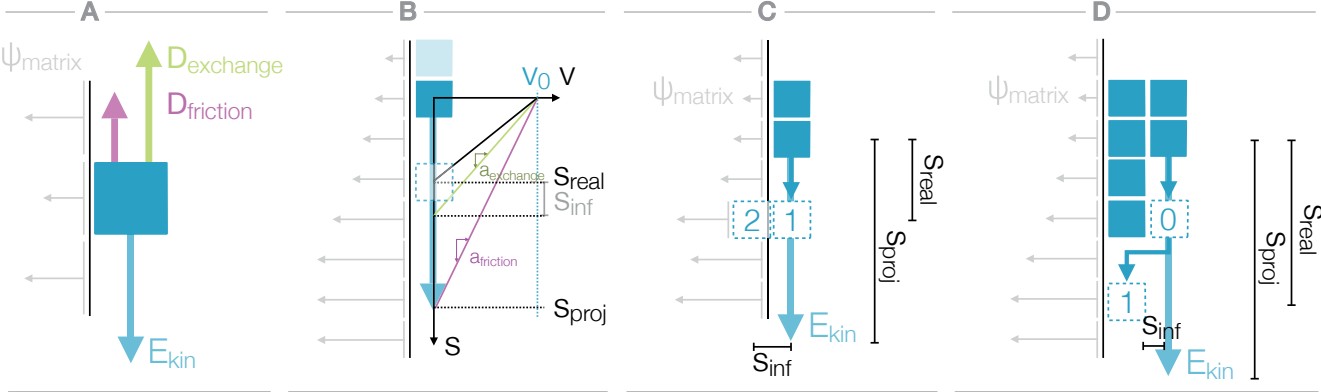

**Figure 3.** Macropore flow concept. A: Concept of a water particle at the pore wall possessing a kinetic energy $E_{kin}$ which is dissipated by friction in the macropore network and exchange with the matrix due to the matric potential $\psi_{\mathrm{matrix}}$. B: Projected advection of a particle where the potential advective velocity $v_0$ is decelerated by the $a_{\mathrm{friction}}$ and $a_{\mathrm{exchange}}$ it experiences along the projected path $s_{proj}$ resulting in a reduced step length $s_{real}$. C: Reduced advection with macropore-matrix exchange (1), and possible infiltration $s_{inf}$ (2). D: Fast advection of a particle as film flow to the end of the film (0) and further decelerated advection (1).

15  and Germann, 1981). The drawback of these approaches is that neither the exchange length nor the leakage parameter are observable, and depend on model grid size and on event characteristics (Köhne et al., 2009a).

Here we propose a thermodynamic approach for describing this key process on a physical basis without introducing additional parameters based on the Bernoulli equation:

$$\underbrace{0.5\varrho\,\overbrace{v_{adv}^2}^{const.}}_{E_{kin}} + \underbrace{\varrho g z}_{E_{pot}} + \overbrace{p}^{=0} + \varepsilon_{\mathrm{friction}} = const. \tag{5}$$

Measured advective flow velocity in earthworm pores range closely around $7.5 \times 10^{-2}\,\mathrm{m\,s^{-1}}$, as given in table 1. These measurements compare with a theoretical laminar flow velocity through a pipe of the same cross-section and with unit pressure

5  gradient with a factor of about 500.

This theoretical flow velocity $u_{mx}$ can be calculated after Hagen-Poiseuille:

$$u_{mx} = 2 \cdot \frac{\rho g R^2}{8 \cdot \eta} \tag{6}$$

with $\rho$ and $\eta$ as density and dynamic viscosity of water, $g$ the gravitational acceleration, and $R$ as radius of the pore. Given its velocity, each particle in motion possesses an $E_{kin}$

10  $$E_{kin} = 0.5 m_{\mathrm{particle}} u_{mx}^2 \tag{7}$$





| advective velocity $\mathrm{m\,s^{-1}}$ | mean of n trials | source |
|---|---|---|
| $7.2 \times 10^{-2}$ | 27 | Shipitalo and Butt (1999) |
| $5.6 \times 10^{-2}$ | 29 | Shipitalo and Butt (1999) |
| $7.7 \times 10^{-2}$ | 16 | Weiler (2001) |
| $5.8 \times 10^{-2}$ | 12 | Zehe (1999) |
| $10.2 \times 10^{-2}$ | 53 | Bouma et al. (1982); Wang et al. (1994) in Weiler (2001) |
| $3.8 \times 10^{1}$ | | after Hagen-Poiseuille |

**Table 1.** Measured mean maximum advective velocity in burrows of the earthworm *Lubricus terrestris* at a mean radius of $4.5\,\mathrm{mm}$ and theoretical value after Hagen-Poiseuille.

With this and the current velocity of a particle $u_{real}$, we may estimate the dissipation by friction in the macropore $\varepsilon_{\text{friction}}$ as impulse $I_{\text{friction}}$ counteracting the hypothetical $E_{kin}$ with:

$$I_{\text{friction}} = E_{kin}/u_{real}. \tag{8}$$

Following Kleidon and Schymanski (2008) and Zehe et al. (2013) soil water experiences a certain capacitative (or capillary binding) energy density $dE_{cap} = \Psi dV_\theta$, as matric potential is a negative energy density. Wetting and drying due to macropore-matrix exchange affects its capillary binding energy approximately as:

$$\varepsilon_{\text{exchange}} = dE_{cap} = \varrho g \frac{\partial \Psi_z}{\partial \theta_z} \cdot \theta d\theta \tag{9}$$

with $\Psi_z$ as matric pressure head in a certain depth $z$ and $\theta_z$ as volumetric soil water content. With this we can estimate dissipation $\varepsilon_{\text{exchange}}$ during the infiltration of one particle as impulse by using the particle volume $V_{\text{particle}}$ and a projected infiltration flux $q_{\text{exchange}}$:

$$I_{\text{exchange}} = \varrho g \frac{\partial \Psi_z}{\partial \theta_z} \frac{V_{\text{particle}}}{q_{\text{exchange}}} \tag{10}$$

The projected infiltration rate $q_{exchange}$ is calculated as Darcy flux $q_{exchange} = k_u(\psi) \cdot -\psi/2r_{\text{particle}}$. Notice that this is only the necessary assumption for the change of $\theta$ in equation 10 directly at the interface. All state-dependent variables are formulated as geometric mean of the references at initial depth $z_i$ and projected depth $z_{proj}$ in a predictor corrector scheme.

Now, the reduced advective velocity of a particle is estimated using friction and exchange drag acting against $E_{kin}$ of the particle in steady state:

$$u_x = -\frac{E_{kin}}{I_{\text{exchange}} + I_{\text{friction}}} \tag{11}$$





If the projected infiltration exceeds the particle radius $q_{\text{exchange}} \cdot \Delta t > r_{\text{particle}}$ the particle will be transferred to the adjoining matrix. With the given equations, the dynamic film flow and infiltration into the matrix is governed by the state-dependent retention properties of the soil (van Genuchten parameters) and the supply of new particles.

### 3.4 Infiltration into macropores and the matrix domain at the upper boundary

With the extension of the model to two dimensions the partitioning of infiltration into macropores and soil matrix became became an important aspect of the model. As pointed out by Weiler (2005); Nimmo (2011) and others, initialisation of the macropores is critical and non-trivial. We employ a generalisation of the concept of macropore drainage areas (Weiler, 2005; Weiler and Naef, 2003) and the concept of preferential flow initiation and partitioning after Nimmo (2011): Precipitation is converted into particles. They are randomly distributed over the top boundary. All particles which happen to fall on soil, first form a film layer similar to the macropore walls described earlier. Excess precipitation or particles directly falling on macropores are redistributed to the macropores according to proximity and capacity. If one macropore's capacity is reached, it is excluded from the redistribution process. Particles in the film layer are included in the diffusive calculation step of the top matrix cells. Particles in the macropore domain are treated as film flow advection and possible infiltration from the macropores into the matrix as described above. Thus, infiltration is only limited by the transport capacity of matrix and macropores. The higher the soil matrix infiltration capacity the lower the share of particles entering the macropores.

### 3.5 Data requirement, technical implementation and numerical issues

The parameterisation of the echoRD model based on observables is a key objective of this study. As pointed out previously, the required parameters for the model are retention characteristics (van Genuchten parameters) and a lateral and vertical density distribution of macropores. The retention properties of the soil matrix can be measured in standard pedophysical analyses.

To derive macropore density distributions horizontal panes of dye tracer stains (e.g. Brilliant Blue experiments) can be analysed with the model preprocessor. With this we make use of experimental data directly as explained in Appendix B. Moreover initial soil water content and a precipitation time series need to be defined.

We rely on sequential calculation of the process domains:

1. infiltration at the top boundary into matrix and macropores,

2. diffusive matrix flux as spatially explicit 2D random walk,

3. film flow in the macropore,

4. macropore-matrix interaction (infiltration and exfiltration).

Checks for saturation and percolation below the lower boundary are performed after step 2 and 3. The time step is controlled through a Courant and Neumann criterion based on the maximum possible diffusive and advective step at the current $max(\theta_t)$





| Parameter | $k_s$ | $\theta_s$ | $\theta_r$ | $\alpha$ | $n$ |
|---|---|---|---|---|---|
| | $\mathrm{m\,s^{-1}}$ | $\mathrm{m^3\,m^{-3}}$ | $\mathrm{m^3\,m^{-3}}$ | $\mathrm{m^{-1}}$ | - |
| Name | $\mathrm{x10^{-6}}$ | | | | |
| Loamy Sand | 16.97 | 0.401 | 0.035 | 11.5 | 1.47 |
| Silty Loam | 3.667 | 0.486 | 0.015 | 4.8 | 1.21 |
| Loess$^W$ | 2.324 | 0.475 | 0.025 | 1.94 | 1.21 |
| Weiher mean | 4.27 | 0.44 | 0.07 | 2.75 | 1.25 |
| Weiher std | 2.7 | 0.03 | 0.04 | 1.97 | 0.08 |

**Table 2.** Soil matrix retention parameters used in the application tests. Loamy Sand and Silty Loam after Carsel and Parrish (1988). Loess$^W$ refers to measured values from soils at the spot of the experiment sect. 4.2.
Weiher comprises seven ensemble soil matrix references of the Weiherbach basin as mean and standard deviation.

or occupied bin respectively:

$$\Delta t_D = \Delta z^2 / 6 max(D_{max(\theta_t)}) \qquad \text{and}$$

$$\Delta t_u = \Delta z / max(k_{max(\theta_t)}) \tag{12}$$

With regard to the representative domain, the interrelation of particle size and the numerical grid is noteworthy. The desired resolution and stochastic stability of the model is controlled by the grid size and the number of particles which represent saturation. Both are required model parameters. Obviously, this quickly leads to a large number of particles if we seek to resolve processes which locally change soil moisture by few percent only. The following tests are realised with a relatively fine
grid and a relatively large number of particles to avoid instabilities and artefacts.

## 4 Model application tests and experimental references

In this section, we outline our application tests of the echoRD model and a reference to real-world conditions in order to examine the capability of the chosen simplifications. In order to focus on the proposed concept and hypothesised process descriptions, the following tests are realised with an underlying grid resolution for particle density calculation of $5\,\mathrm{mm}$. The
10 water particles are set to a size of 0.002 times a grid cell (equivalent to $0.33\,\mathrm{mg}$).

With the extension to two dimensions and the introduction of representative macropores, the test applications shall especially address the following aspects:

**a)** 2D diffusive, non-uniform soil water redistribution

**b)** interaction of 1D advective paths with 2D soil matrix

**c)** sensitivity to state variables and model parameters





**d)** robustness of the representative macropore setting

**e)** reproduction of a real-world irrigation experiment

### 4.1 Generic application testcases

The central benchmark of the model is a series of generic test applications with different soil types, precipitation intensities and antecedent soil moisture. The aim is to examine the consistency and capability of the model and the self-controlled non-uniform flow with regard to points a–c. The test matrix is spanned by:

- soil water retention parameters for a sandy soil, a loamy soil and a loess soil (Tab. 2),

  - two different antecedent moisture states at $0.15\,\mathrm{m^3\,m^{-3}}$ and $0.31\,\mathrm{m^3\,m^{-3}}$, and

  - precipitation intensities at $10$, $40$ and $60\,\mathrm{mm\,h^{-1}}$ lasting for $30\,\mathrm{min}$

The resulting model runs are compared visually based on the infiltration patterns and numerically based on the distribution of newly added particles. Additionally, we compare the resulting travel depth distributions based by means of the first three central

moments. In these scenarios, the macropore network is the same. It is defined based on earth worm macropore assessments in an agricultural loess landscape using the preprocessor (Appendix B). To gain insight into the model robustness, alternative definitions of macropores based on the same input statistics are compared separately (aspect d). Moreover we test the influence of different particle resolutions with $100$, $200$ and $500$ particles per grid cell at $\theta_s$ for some examples.

### 4.2 Plot-scale irrigation experiment as real-world testcase

We conducted a series of plot-scale irrigation experiments in different soil landscapes (Jackisch, 2015). Our model development is founded on these findings, based on the hypothesis that irrigation experiments can reveal the distribution of advective flow paths and the resulting non-uniform soil water redistribution characteristics (Jackisch et al., 2017). By using a sprinkler with a very fine drop spectrum and a drip irrigation pad in the presented case on undisturbed surface conditions, we neglect drop

splash impact (Iserloh et al., 2013) and macropore drainage area connectivity (Weiler and Naef, 2003). Diffusive soil water transport parameters are determined based on laboratory analyses of undisturbed soil cores for their retention properties.

Because the model is intended as exploration tool extending real world experiments, a further test of the model aims at reproducing one experiment in the Weiherbach basin in South-west Germany with Loess soils on a fallow plot ($49.135\,17°$ N, $8.744\,15°$ E, Oct. 20, 2015). The irrigation was realised with $40\,\mathrm{mm}$ water in $2\,\mathrm{h}$ on a $1\,\mathrm{m^2}$ plot with a drip irrigation pad. The

water was enriched with $5\,\mathrm{g\,l^{-1}}$ potassium bromide (KBr) salt tracer and $4\,\mathrm{g\,l^{-1}}$ Brilliant Blue dye tracer. The plot remained covered during the whole experiment until excavation. The state was monitored with a TDR soil moisture tube probe (Trime IPH, IMKO GmbH) and time-lapse 3-D ground penetrating radar. The plot was excavated $20\,\mathrm{h}$ after irrigation on set for dye stain recovery (Fig. 4). In addition two core samples ($80\,\mathrm{mm}$ diameter) have been drawn $20\,\mathrm{h}$ and $30\,\mathrm{h}$ after irrigation onset, respectively. The cores have been sliced each $15\,\mathrm{mm}$ and were analysed for Bromide concentration as in (Jackisch, 2015).



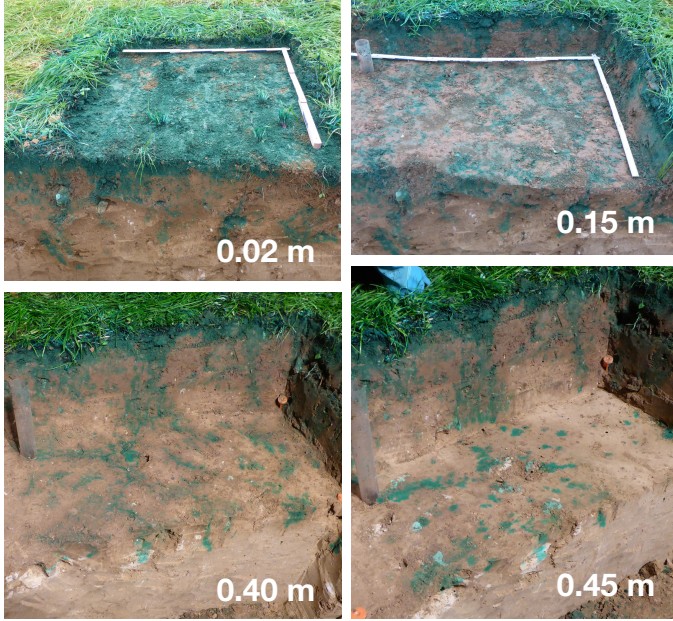

**Figure 4.** Weiherbach irrigation experiment as model reference. Brilliant blue dye stains in excavation horizons.

The echoRD model setup is based on a stochastic matrix definition of seven equally valid ensembles of measurement and literature references (Zehe, 1999; Zehe et al., 2001; Plate and Zehe, 2008, see Tab. 2 Weiher). The macropore domain has been parameterised based on observed dye stain patterns in four depth layers using the preprocessor (Appendix B).

## 5 Results

### 5.1 Generic application tests

The generic application tests show the capability of the model to calculate self-controlled, non-uniform infiltration patterns (Fig. 5 and 6).

The simulations of $40\,\mathrm{mm}$ irrigation in $0.5\,\mathrm{h}$ on loess silt with different antecedent soil water content show the development of a non-uniform flow field conditioned by the representative macropores (Fig. 5). The overall soil water dynamics (left panels) exhibit a quickly expanding advection in the larger macropores. The respective breakthrough curves (marginals and right panels) allow to quantify this behaviour. Already after $10\,\mathrm{min}$ new particles reach a depth up to $0.2\,\mathrm{m}$ while the center of mass is around $0.05\,\mathrm{m}$. The results also show that the fast advective displacement requires continuous supply. After the end of irrigation soil water is mostly redistributed diffusively which can be seen as blur in the soil water content. This is also depicted by relatively steady breakthrough curves. Thus the model proves capable to simulate advective film flow, macropore-matrix exchange and 2D diffusive redistribution dissipating the lateral gradients. This proves aspects a and b.



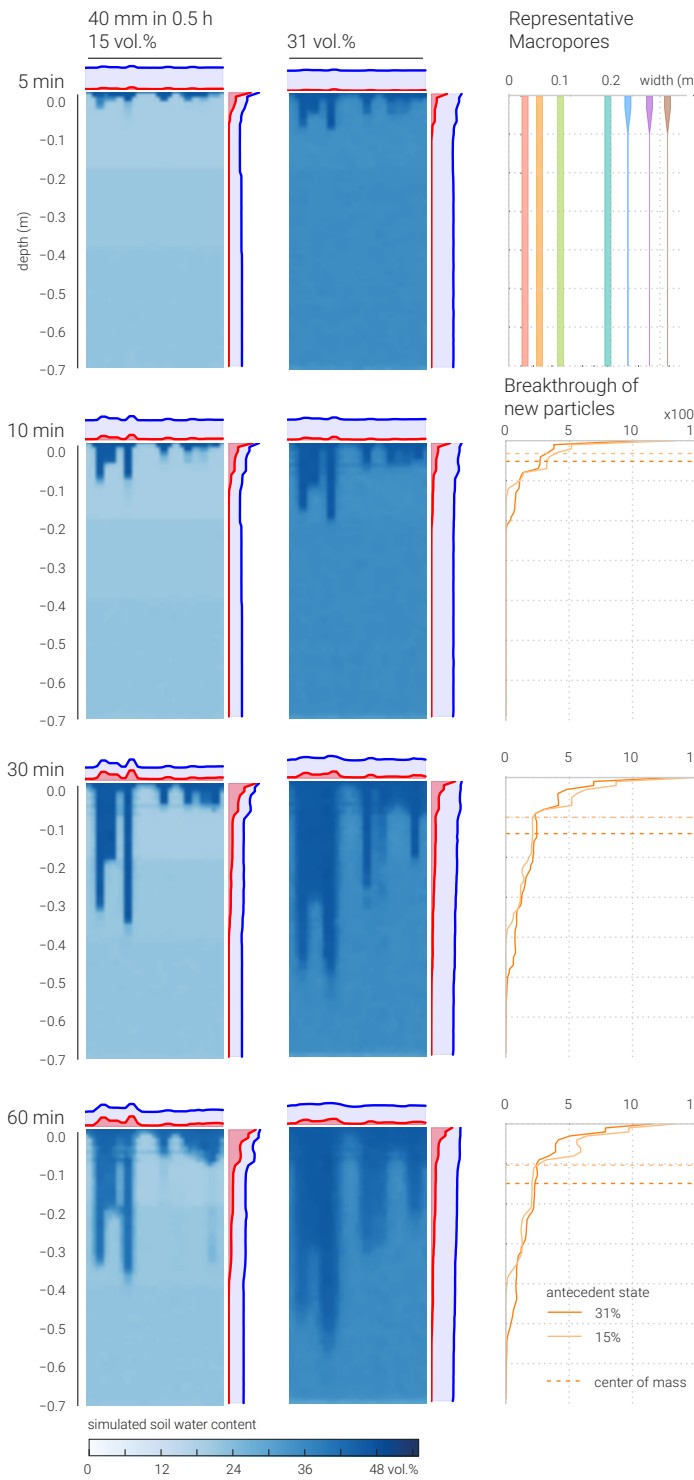

**Figure 5.** Simulated soil moisture dynamics in generic application tests of Loess soil. Marginals give the distribution of all particles (blue) and newly infiltrated particles (red). In the right column the representative macropore domain definition and the breakthrough curves of new particles at the different time steps are given.



Comparing the different soil types loamy sand, silty loam and loess silt, the two respective antecedent moisture states and three irrigation intensities, more insight into the simulated soil water dynamics is given (Fig. 6). Generally, with increased supply intensity the non-uniform flow field becomes more prominent. However, also moderate intensity can develop such patterns, depending on the diffusive momentum. It becomes apparent that the more conductive the matrix the less pronounced the advective fraction became. The diffusive redistribution of particles is especially obvious for the highly conductive loamy sand. With low supply intensities and high antecedent soil water content, this leads to almost uniform infiltration. The diffusive redistribution is especially visible when comparing the results of different antecedent states. Under dry conditions the film flow is experiencing more drag with less exfiltration into the matrix. Wet conditions and more conductive soils lead to less friction but also more lateral displacement. In the long simulation runs (bottom row) the lateral gradients are increasingly dissipated as one would expect.

Moreover, a larger supply is sustaining the advective fraction, the greater the reached depth or breakthrough. When analysing the simulated dynamics this also led to different apparent velocities in the respective macropores (see supplement video). This behaviour is consistent with field observations and our expectations. As such, the model proves to fulfil the required objectives a–c.

A more quantitative reference is obtained when comparing the depth distribution of new particles of the application tests directly (Fig. 7). The temporal dynamics of the infiltration patterns in loamy sand start with a largely intensity controlled situation (low deviation between antecedent conditions). The picture changes to antecedent state controlled top soil retention for the higher intensities with very similar profiles respectively. Total irrigation amount controls deeper percolation in the later course of the simulation. There, the deeper tailing is reduced by the top soil retention leading to different reached depths of all simulations with high irrigation intensity. Low intensities resulted in similar overall breakthrough. In Appendix Fig. D1 the breakthrough curves after 1 h simulation of all generic application tests show the same dependency on soil type and antecedent state.

It is noteworthy to regard the development of the corresponding moments of the depth distribution of infiltrating particles (Fig. 8). The average travel depth increases with time in a clearly non-linear way during rainfall driven conditions, and remains nearly constant during non driven conditions afterwards. The variance exhibits a similar temporal pattern. The skewness of the travel depth distributions generally peaks shortly after the irrigation onset and decreases after that. This rising limb and the early peak marks the initial development of "flow fingers" in single or few macropores. The activation of additional macropores does however reduce the skewness as the median of travel depth starts to "chase" the mean. This finding shows clearly that a flow pattern that is strongly dominated by preferential must not necessarily be skewed (Dreuzy et al., 2012). As the third moment tends to minimise for the cases with high antecedent soil moisture and thus lateral diffusion, the qualitative observations of relatively smooth infiltration patterns in Fig. 6 are reflected very well. The temporal evolution of the dispersivity in panel D reveals clearly that the transport is not well mixed during the entire duration of the rainfall forcing. It operates in the near field, as the the variance grow quadratically with time. Later, diffusion dominates the soil water redistribution.

In addition, we performed model parameter-related tests drawing different realisations of the macropore setting from the same ensemble (Fig. 7 right panels). The breakthrough curves of 8 alternative realisations of the representative macropores





**Figure 6.** Table of simulated soil moisture dynamics in generic application tests for Loess. Marginals give the distribution of all particles (blue) and newly infiltrated particles (red).





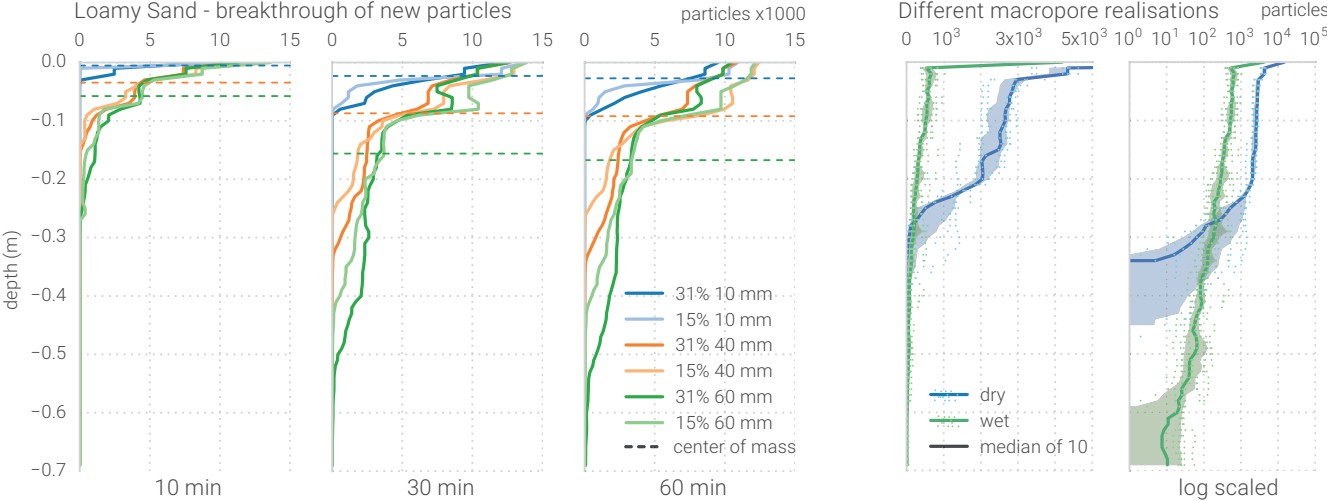

**Figure 7.** Simulated depth distribution of new particles in generic application tests. Left: Loamy Sand at different times after irrigation start. Right: Simulations with different macropore realisations based on the same input data.

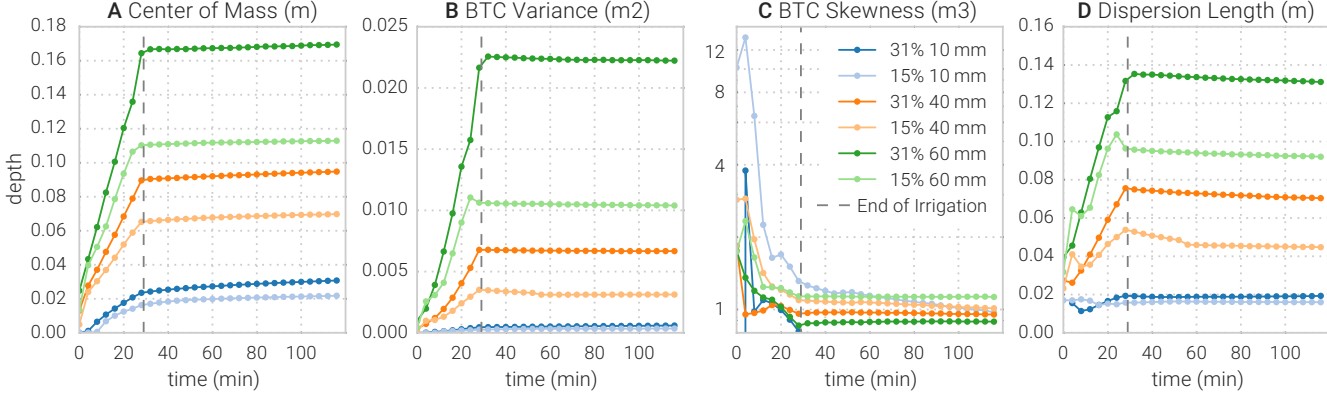

**Figure 8.** Dynamics of the moments of the breakthrough curves of loamy sand simulations. A: first moment as center of mass, B: second moment as variance of depth distribution, C: third moment as skewness of depth distribution, D: dispersion length defined by $<A>/<B>$.

under two different antecedent conditions are given with the 0.25 and 0.75 percentiles as variability bands. In order to evaluate the effect on potential contaminant breakthrough, a log-transformed plot is given. The results are within realistic bands and well below the uncertainty of tracer recovery of such experiments. Thus test aspect d is achieved. Variance can be narrowed by defining a larger domain width. This may become important for highly skewed macropore distributions, where the requirement for the minimal domain width may be higher than assumed.



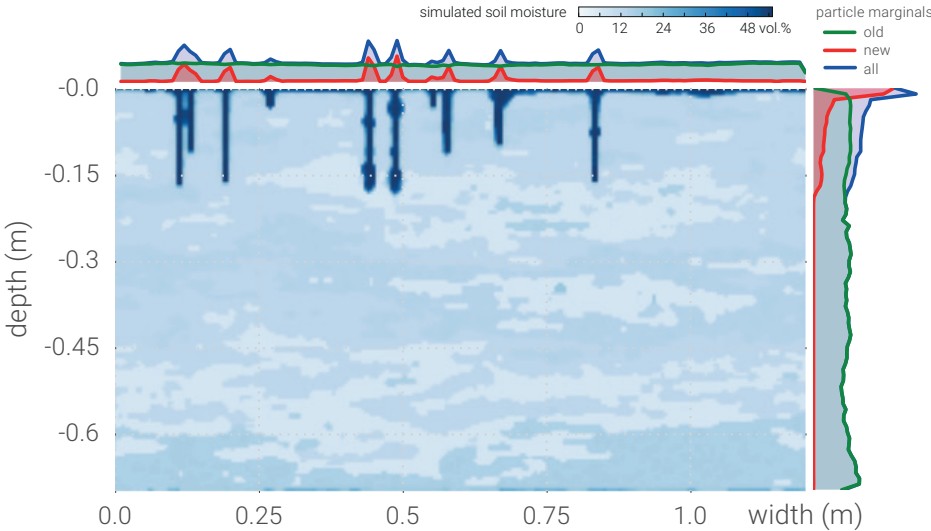

**Figure 9.** Simulated particle distribution in mimicry of the Weiherbach. 40 minutes after irrigation onset. A movie of the simulated dynamics is given as supplement.

Tests with different particle resolutions showed that too coarse definitions can result in local averaging, which underestimates the actual depth distribution of the infiltrating water. A similar effect was observed with very coarse internal calculation grid definitions, which could no longer represent local state changes due to infiltration from the macropores into the surrounding matrix.

## 5.2 Reproduction of irrigation experiment

The last benchmark is the reproduction of observed tracer profiles based on measured parameters (test aspect e).

The simulation depicts the observed stain patterns and concentration profiles very well (Fig. 9, video as supplement). Despite a lack of precise observation of the actual non-uniform flow dynamics, the simulated behaviour also matches the time-lapse ground penetrating radar records. Moreover, the simulation snapshot taken at about one third of the irrigation period refers well to the profiles of tracer and soil water content recorded in the field (Fig. 10). For comparability, the simulated distribution of new water particles is converted to a tracer mass by assuming a domain thickness of one particle diameter, referring the simulated mass to the sampled volume and applying the $Br^-$ concentration in the irrigation solution. Moreover, the snapshot is scaled to the total irrigation to be conclusively comparable to the recovered profile. Despite overall good fit, the profile still deviates at shallow accumulation around $0.05\,\mathrm{m}$ and at deeper percolation to $0.3$ to $0.4\,\mathrm{m}$. This is very much in line with the findings in the generic application tests presented earlier. The overall shape of the distribution of new particles established relatively soon after irrigation onset while the fast and slow fractions are fixated after irrigation end.

Changes in soil water content are accumulated to the integration volume of the TDR sensor for better comparison (Fig. 10, right panel). The simulation fits between the reference records at 28 and $60\,\mathrm{min}$ after irrigation onset. While the overall shape



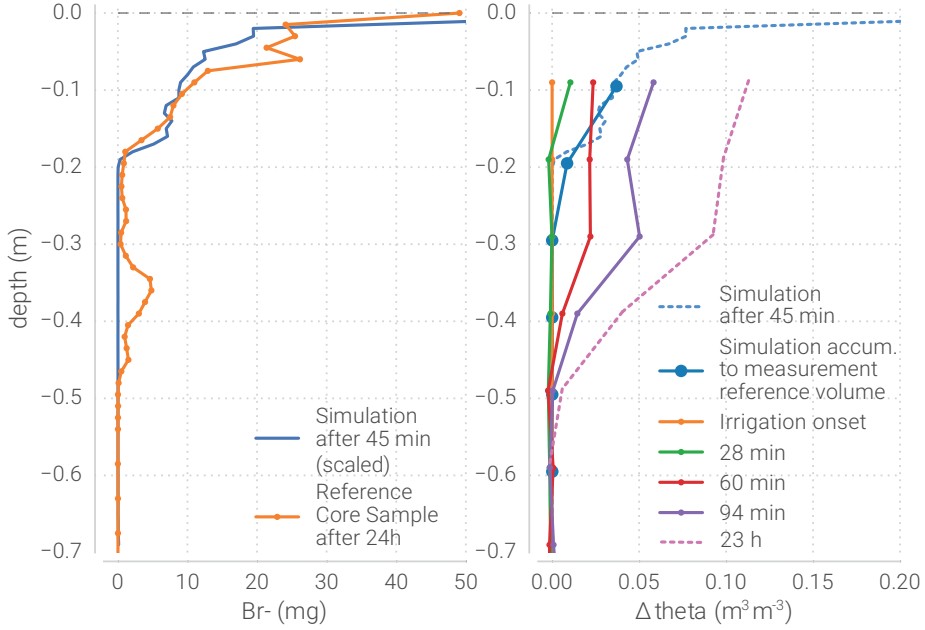

**Figure 10.** Simulated and observed tracer (left) and soil moisture profiles (right) in irrigation experiment. Tracer mass scaled to core sample volume and total irrigation after 2 h.

of the profile is plausible, the high water content near the surface is not reflected in the early soil moisture measurements. It is noteworthy, that by the large integration volume of the sensor much of the characteristics of the profile is strongly smoothed out.

A closer look at the outcrops in Fig. 4 exhibits a deviation of the wetting front and the stain pattern which hints to chromatographic effects due to a shift in flow velocities switching from high velocities during well-supplied states near saturation to an purely diffusive transport. This process is represented in the model too: The flow in the macropores takes place at different

5   velocities until very shortly after irrigation end. Then diffusive redistribution alone governs the lateral water transport. Similar results have been found in Brilliant Blue tracer experiments and simulations with the same model by Reck et al. (in review).

Moreover, it can be noted that the modelled depth distribution of new particles coincides with the observed tracer breakthrough. This is especially interesting because the macropores are defined to reach through the full domain as earthworm burrows are reported to reach depths below 2 m. Hence the self-controlled limitation of advective flow in the macropores

10   appears to be capable to reproduce the true process.





## 6 Discussion

### 6.1 Model adequacy

The general adequacy of the echoRD model to represent non-uniform irrigation water redistribution is outlined by the generic application tests. The water particles move realistically in the conjugated domains under the tested conditions. Also the mimicry of irrigation experiment based on directly measurable parameters corroborates the proposed model framework with regard to structural adequacy (Gupta et al., 2012; Gupta and Nearing, 2014) and the intended objectives. Further testing should explore the model capabilities under various macropore settings in heterogeneous soils. Especially, the universality of the proposed macropore-matrix exchange concept deserves further assessment.

During the development we followed Clark et al. (2011) by testing multiple alternative working process hypotheses for a) initial irrigation water redistribution, b) initial advective velocity reference, c) macropore-matrix exchange, and d) macropore film flow as further detailed in Jackisch (2015). During preliminary testings the here presented set performed most realistically. However, we encourage further testings and development of more hypotheses within the framework. Especially since the Lagrangian method using water itself as particles required to abandon most of the well-established theories of soil water movement in an Eulerian domain, there is ample room for further adaptations, extensions and even falsification of the proposed ideas. The provided repository of the model shall invite and prepare the community for that.

### 6.2 Representative structured domain and particle concept

Building on the idea of self-similarity in flow networks going back to the works of Rodriguez-Iturbe and Rinaldo (1997); Rinaldo et al. (2014) we propose a topologically explicitly structured domain setup for the plot scale. The presence and importance of interfaces in soils (among others Hassanizadeh and Gray, 1990; Lehmann et al., 2012) led to the proposition of the combination of a 2D matrix, which accounts for non-equilibrium lateral and vertical diffusion, and multiple 1D vertically oriented advective structures, which account for fast vertical redistribution. With this, we also seek to combine some of the existing modelling approaches to multi-phase soil water dynamics (as introduced Jury and Roth, 1990; Vogel and Roth, 2003; Gerke, 2006; Vogel et al., 2006; Sander and Gerke, 2009).

We explicitly avoid a direct and tortuous representation of a macropore network as commonly observed (e.g. for earthworm burrow systems Capowiez et al., 2003, 2011). All effects connected to friction in the macropore (which includes the inclination of the macropore gallery and pore roughness) are implicitly summed up in equation 8. When more information is given, this can be further differentiated in a future adaptation. The effect of coatings in earthworm burrows (Jarvis, 2007) was so far neglected due to a lack of experimental references. However, a dynamic coating factor is foreseen to scale the macropore-matrix interface. Although most of the specific references have been drawn with relation to earthworm burrows, the current concept is intended to apply to any kind of macropore.

The combination of the particle approach with the connected domains avoids a number of implicit assumptions for the exchange between the domains. Our energy-balance approach to film flow in the macropores enables analyses of different infiltration patterns with self-controlled advection and diffusion. In addition to this process hypothesis, many alternative ap-




proaches to model the interfacial processes and the behaviour within the respective domains can be imagined. For this, the

30  echoRD model allows for direct process hypothesis testing with the same objects.

We have shown that different infiltration patterns emerge based on different antecedent conditions and forcing of the representative structured domain (Sect. 5.1). The influence of different realisations of the representative macropore domain from the same ensemble has been small. This does corroborate the validity of the selection of the representative domain.

The non-stationary and non-linear dispersivity underpins the limitations when the processes during driven conditions are subsumed by explicit and universal parameterisation. However, diffusive transport dominates quickly after the supply ceases. This motivates a potential use of the full echoRD model to derive state- and forcing-dependent distribution functions for the

advective flow field, which can successively be used in more simple versions of the particle model like our 1D approach (Zehe and Jackisch, 2016) or the MIPs model (Davies et al., 2011). Moreover, the concept can also be downscaled to analyse pore water fractionation in the vadose zone extending our initial binning approach in the pore spectrum. Both aspects take place within the same framework opening possible ways to bridge the different scales.

### 6.3  Capability and limits of the model

Although the echoRD model posses many degrees of freedom to adjust its behaviour, it is not intended for parameter fitting. Instead, the model is proposed as exploration tool capable to extend real world experiments. As such, the model requires very few parameters, which can all be derived from suitable experiments: Soil matrix parameters are used for determination of the diffusive and storage properties of the soil and consist of soil water retention parameters. If desired, the van Genuchten model can be replaced by any other soil water retention model. Each calculation grid of the matrix domain can be assigned to

a different soil matrix definition. Macropores host the advective flow and are determined by the spatial distributions (relative distances and depth) and a reference to maximum flow capacity. In addition some coating factor may be defined for earthworm burrow coatings which scales the contact interface to the matrix.

There has been much debate about the derivation of effective parameters in hydrological models (e.g. Bashford et al., 2002; Neuweiler and Vogel, 2007). With the physical description of the two domains and their exchange, the parameters become

much more specific and scale-aware. Soil water retention properties are determined for the matrix in standard lab procedures while macropore settings can be quickly assessed with dye staining experiments in the field (e.g. Reck et al., in review). With this we also aim to contribute to model falsifiability (e.g. Harte, 2002). As it is making direct use of the laboriously gathered and valuable data from experiments, surveys and monitoring it also improves the matching of model concepts and hydrological observables Beven (1993, 2006).

We envisage further use with dynamic macropore settings as the domain may update once it is empty and as foundation to derive state and forcing dependent stochastic site properties which can be used in more lumped versions of the approach. Since the particle domain can always be converted into an Eulerian field of matric potential or soil water content and vice versa the model can also be linked to a Richards model for periods when the diffusive flow assumption is valid.





In the application tests it was seen that the model is quite sensitive to antecedent conditions. Under uncertain state data this
may lead to susceptibility of the model to uncertainty about the macropore-matrix exchange which can be amplified through
the non-linear retention properties. Moreover, the model has shown sensitivity to dead ended macropores. Hence special care
has to be given to provide valid data on the macropore distribution.

## 6.4   Numerical concerns

The simulation of soil water dynamics based on water itself as particles is generally very different from the common particle
tracking for solutes. On the one hand there is no external drift and the activity of each particle depends on its neighbours.
On the other hand a very large number of particles is needed to enable robust calculation of the low event signal against a
rather high background or pre-event concentration. The reason for this is that the resolution of the process dynamics scales
with the number of particles per volume reference (grid cell in our case). At the same time do we require relatively small
volume references to avoid integration over too large scales. All of these points demand a large number of particles which
require frequent state update about their relative concentration distributions and binning in the pore water space. Moreover, the
calculation of film flow with many particles is similarly self-depending.

The Courant and Neumann criterion for the time-step control are calculating a global specification. Hence local wetting
causes very small time-steps for the whole model. In combination with the previous concerns this makes the model compu-
tationally very expensive. Due to the self-dependent state, we could not find any option to make use of the more efficient
continuous time random walk methodology (Metzler and Klafter, 2000; Delay and Bodin, 2001; Dentz et al., 2012).

At the current state of experimental code the model runs at about 10 to 200 times more slowly than the real time of the
simulated case. Despite its potential, we abandoned trials using grid-free methods to calculate the particle density e.g. by
Voronoi polygon area calculation (Rycroft, 2009) as they multiplied the calculation effort even further. A next step will be to
optimise the model for performance in the frequent state-updates.

## 6.5   Model-based extension of real world experiments

One of the intended uses of the model is to overcome the limitations of destructive irrigation experiments. So far it is impos-
sible to repeat tracer-based plot irrigation experiments as the site needs excavation for sampling. Moreover are the spatial and
temporal scales of such experiments very difficult to observe Jackisch et al. (2017). Since the model is promising with re-
gard to simulate infiltration, advective flow, macropore-matrix exchange and diffusive redistribution without explicit exchange
parameterisation it opens ways to virtual experiments about controls of non-uniform subsurface flow.

Fig. 7 has hinted to the interplay of supply rates and duration for advective flow breakthrough. Jackisch (2015) presented an
initial model analysis of the effectiveness of fast drainage under different antecedent conditions and forcing. The simulations
reveal that preferential flow occurs under all conditions corroborating the findings of Nimmo (2011). Under wet antecedent
conditions also moderate rainfall events can result in substantial break through. The same was shown by Reck et al. (in review)
in tracer experiments and subsequent modelling at different seasons.





Besides the initial development of "flow fingers" and the evolution of the skewness of the depth distribution of the event water (Sec. 5.1), another aspect is that a large number of macropores does not necessarily result in deeper percolation since the irrigation supply is distributed to all effective macropores. This can lead to situations where the supply rates in the macropores

drop below the macropore-matrix exchange rates. As the model is capable to reproduce this behaviour, we hope that it can contribute to unify the debate about the importance of non-uniform flow and preferential flowpaths.

## 7 Conclusions

In the recent paper (Zehe and Jackisch, 2016) we provided the foundation for an alternative representation of soil water diffusion based on a random walk of water particles in the space domain. We showed that this is a true alternative to solvers of the Richards equation. In this study, we extended the approach to a multi-domain model of advection and diffusion in a representative structured domain with a 2D matrix hosting topologically explicit 1D macropores as a physical and least ade-

5 quate representation of the processes – the echoRD model (eco-hydrological particle model based on Representative structured Domains).

In a series of application tests we showed the model's capability to represent a) 2D diffusive, non-uniform soil water redistribution, b) self-controlled interaction of the 1D advective paths with the 2D soil matrix, c) sensitivity to state variables and

10 observable model parameters and d) robustness of the representative macropore setting based on macropore depth distributions. Moreover, the model was successfully used to mimic a real-world irrigation experiment based on measured parameters.

This implies the structural adequacy of the model simulating advective flow as dynamic film flow in topologically explicit macropores and accounting for macropore-matrix exchange based on an energy-balance approach. The multi-domain interplay of advective and diffusive soil water redistribution exhibited a non-linear temporal evolution of the dispersivity. While the

15 process description appears rather sophisticated, its parameterisation is very simple as the model relies on soil water retention properties for the soil matrix and data about the depth distribution of effective macropores.

As the model is intended as learning tool to extend real world experiments we have shown its potential for virtual experiments under different antecedent states, macropore settings and precipitation forcing. The model is also envisaged to deliver a physically based foundation for infiltration statistics which can then inform Markov process of higher order in simpler 1D versions of the model Zehe and Jackisch (2016) scaling the approach to the hillslope by means of definition of representative

5 soil domains connected to an explicit lateral structure (Zehe et al., 2014).

## Acknowledgments

This study contributes to and greatly benefited from the "Catchments As Organized Systems" (CAOS) research unit. We sincerely thank the German Research Foundation (DFG) for funding (FOR 1598, ZE 533/9-1). Especially, we thank Loes van Schaik and Niklas Allroggen for the initial discussion of macropore representation and joint experiments. Moreover, this





10  study greatly benefited from the cooperation with the KIT Engler-Bunte-Institute analysing hundreds of samples for Br⁻ and countless student assistants during the field work and laboratory analyses.

Moreover, the realisation of the many test runs of the model would have been impossible without the HPC infrastructure of the universities of Baden-Württemberg. We are very grateful for this unique opportunity and support. The echoRD model, reference data and the presented testcases are accessible as GitHub repositories: https://github.com/cojacoo/echoRD_model.

## 5  Appendix A: Used Variables

| Symbol | Description | Unit |
| --- | --- | --- |
| $D(\theta)$ | Diffusivity | $\mathrm{m^2\,s^{-1}}$ |
| $E_{kin}$ | Kinetic energy | $\mathrm{kg\,m^2\,s^{-2}}$ |
| $\varepsilon$ | Dissipation | $\mathrm{kg\,m^2\,s^{-2}}$ |
| $\eta$ | Dynamic viscosity of water | $\mathrm{kg\,m^{-1}\,s^{-1}}$ |
| $g$ | Gravitational acceleration | $\mathrm{m\,s^{-1}}$ |
| $I$ | Impulse counteracting $E_{kin}$ | $\mathrm{kg\,m\,s^{-1}}$ |
| $k(\theta)$ | Unsaturated hydraulic conductivity | $\mathrm{m\,s^{-1}}$ |
| $m$ | Mass | $\mathrm{kg}$ |
| $n$ | Count | - |
| $\Psi$ | Matric head | $\mathrm{Pa}$ |
| $\psi$ | Matric head as column water | $\mathrm{m}$ |
| $q$ | Flux | $\mathrm{m\,s^{-1}}$ |
| $R$ | Macropore radius | $\mathrm{m}$ |
| $r_{\text{particle}}$ | Particle radius | $\mathrm{m}$ |
| $\rho$ | Density of water | $\mathrm{kg\,m^{-3}}$ |
| $t$ | Time | $\mathrm{s}$ |
| $\theta$ | Volumetric soil water content | $\mathrm{m^3\,m^{-3}}$ |
| $u$ | Advective velocity in matrix | $\mathrm{m\,s^{-1}}$ |
| $v$ | Advective velocity in structures | $\mathrm{m\,s^{-1}}$ |
| $V$ | Volume | $\mathrm{m^3}$ |
| $x$ | Lateral distance | $\mathrm{m}$ |
| $\xi$ | Uniform random number -1..1 | - |
| $z$ | Depth | $\mathrm{m}$ |





**Figure A1.** Example of preprocessing of stain images, patch identification and statistics and resulting macropore positions in representative domain.

## Appendix B: echoRD model setup and preprocessor

The echoRD model can be setup based on soil water retention data (as a table of van Genuchten parameters) for different soil layers and any sort of information about the macropore distribution. The easiest way is to provide images of horizontal outcrops of dye stain patterns to the preprocessor.





The rectified and cropped images with a defined resolution are read and analysed for stained patches using the scikit-image (van der Walt et al., 2014) and scipy.ndimage packages. To do so the patches are identified by using the watershed image processing in scikit-image (Beucher and Lantuejoul, 2006) based on a sobel transformed difference of the green and blue spectrum of the RGB image. Small patches below a given threshold are discarded. Large patches are assumed to consist of multiple

10    macropores and are broken down by means of watershed segmentation. After removal of clutter the patches are labeled and their geometry is assessed.

In a next step these identified patches are analysed for distribution of topological parameters like total number, distance, size and diameter. Based on the least density among all horizons, the representative domain is scaled so that at least one effective macropore exists in the sparsest case. Thus, the fewer macropores the larger the domain.

15    Subsequently, topological parameters are then resampled on the representative domain by allocating all representative macropores to a certain position on the 2D matrix domain based on the observed lateral distance distribution. Moreover, contact areas are defined, depending on the circumference distribution of the patches.

An example is given in figure A1. The code is included in the repository and initiated by `run_echoRD.preproc_echoRD`.

**Appendix C:  The echoRD repository**

This paper is accompanied by a repository at GitHub where the echoRD model and the presented testcases are made publicly available: https://github.com/cojacoo/echoRD_model. The model is developed and tested based on Python 3.5.2. The examples are given as Jupyter Notebooks and as standalone scripts. The packages NumPy, SciPy, Pandas and Matplotlib are required. The preprocessor requests more specific packages as outlined there.

All software and data is given under GNU General Public License (GPLv3) and Creative Commons License (CC BY-NC-SA 4.0) respectively. This is scientific, experimental code without any warranty nor liability in any case. The code is not fully optimised yet and calculations are computationally demanding. However, you are invited to use, test and expand the model at your own risk. If you do so, please contact the first author and repository owner to keep informed about bugs and modifications.

The repository holds the folder `echoRD` with the model engine and the folder `testcase` with routines controlling the model and several setups and exemplary results. For a quick view, the Jupyter Notebooks can be accessed online from the repository home. If you want to run the model yourself, pls. clone and fork the repository.

**Appendix D:  Further model figures**

Fig. D1 presents breakthrough curves of the different soils used in the generic application tests.

The videos of the modelled evolution of soil water content are given in supplement files.





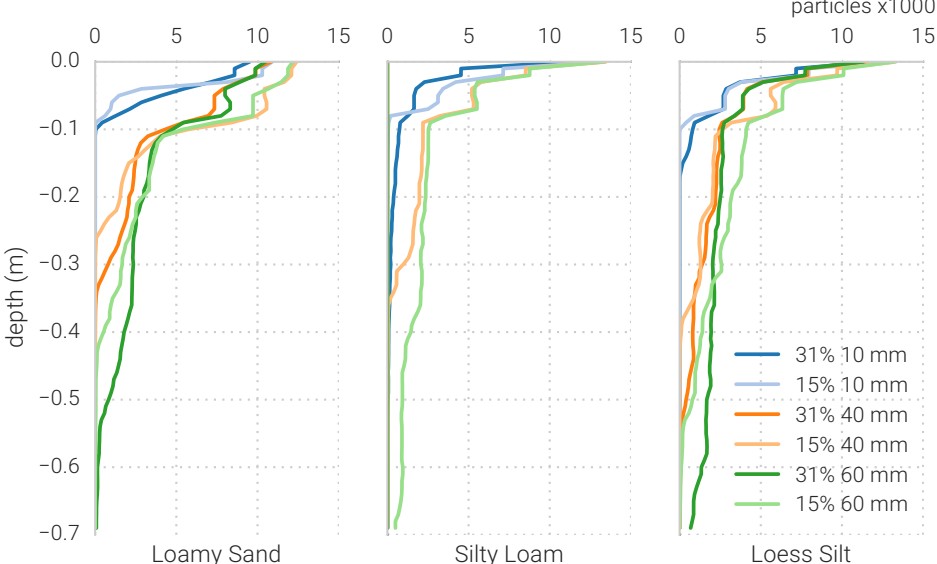

**Figure D1.** Simulated depth distribution of new particles in generic application tests. Different soil types after 1 h.

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
