# Peer review of "Ecohydrological particle model based on representative domains"

_Hydrology and Earth System Sciences, 2017_

## Referee Comment (RC1) · Anonymous Referee #1 · 4 Jan 2018

I do like its originality and the proposed alternative in modelling water movement in unsaturated soils. The authors provide a clear and fair presentation of their model. The paper is well written and well-illustrated.

My comments and suggestions: It is assumed that solutes are moving like water to define the macropores. This is not obvious, since solute can diffuse in existing saturated dead end pores as it is often the case in unsaturated transport. Please comment on this. Is the methodology restricted to 1d vertical macropores? It seems to me that it could be extended to single macropores that have more complicated geometries. The drift term (velocity) is not trivial to me. It is uniform inside a macropore when saturation is reached, it is not before saturation. How is the derivative of the diffusivity handled numerically with macro-pore – matrix interactions? p. 7, line 16: Where is the 0.7 per-

centile coming from ? Solutes are injected at high concentrations (5g/l KBr). Density effects may affect the fluid velocity. I did not understood how the particle breakthrough is computed over the domain. Is it an arithmetic average ? Flux averaged ? This holds for water and contaminant BTCs. The authors are slightly too enthusiastic by interpreting the simulation of the irrigation experiment. First, they provide a comparison for short time (infiltration over 20cm). At this time, it is difficult to identify biases. Second, there is a preferable flow which transported the tracer to a depth of 30-40 cm (see fig. 10) and which is not reproduced by the model. Despite these differences, I agree with the authors that their alternative model is able to simulate that experiment. Concerning the comparison with TDR measurements, how is the support volume defined and how is it taken into account in the modelling?

Typos p. 7, L 28, 'in order' instead of 'in oder'. p. 11, L2-3, 'became' is repeated. Fig. 8, Panel D: Should be' dispersivity' instead of 'Dispersion length'. Dispersion length always increases with time.

---

## Referee Comment (RC2) · Anonymous Referee #2 · 18 Jan 2018

The manuscript deals with an extension of the one-dimensional particle based water transport model of the same authors published in HESS in 2016 to two dimensions with diffuse water flow in the matrix. The model is based on a Langevin equation for "saturation particles" that is equivalent to the Richards equation in saturation form. It accounts for macropore flow and exchange with the soil matrix. The approach is original and provides some interesting features. However, I have some rather conceptual comments regarding the particle rules used to model different flow regimes and the conceptual use of "water particles". This is outlined below.

Comments:

p. 6/7: The authors introduce here a relaxation time to LTE for a local non-equilibrium configuration of particles. What is the correspondence of this rule in a Eulerian frame,

i.e., on the level of the Richards equation? Is it a type of first-order relaxation relation as used by Hassanizadeh and Gray for example?

Eq. (7): What is the mass of a water particle? The "water particles" are merely a conceptual picture, in fact, they correspond rather to "saturation particles". Equivalence between Richards equation in saturation form and the Langevin equation is achieved in the limit of infinite number of particles. The authors should clarify these points. Also the assignation of a particle radius (how is this radius determined) to what conceptually is a point particle is unconventional.

Section 3.3.3 How do the rules established for the Macropore-Matrix interaction correspond to the dual porosity models by Gerke and van Genuchten, for example?

Sections 4.2 and 5.1: The authors refer here to generic application tests as benchmarks. It is not clear however against which benchmarks the model results are compared, or in other words, which are the benchmarks? I could imagine a 2D numerical solution of the Richards equation, for example.

---

## Author Comment (AC1) · 1 Feb 2018

We sincerely thank referee 1 for the constructive comments. With this reply, we respond to these as part of the interactive discussion process. A response addressing the revision of the manuscript in detail will be given after the discussion phase.

*It is assumed that solutes are moving like water to define the macropores. This is not obvious, since solute can diffuse in existing saturated dead end pores as it is often the case in unsaturated transport. Please comment on this.*

The referee is correct that the model cannot account for diffusive solute transport at the current state. Because we consider non-uniform flow during and shortly after events as most critical, the model has a clear focus on this initial phase with high potential

for advection. The time scales for diffusive solute transport are generally much longer. At the same time, diffusive mixing of solute mass among the particles is planned for future versions, which is important when going to long time simulations. With regard to dye staining and tracer experiments an accordance of the solute redistribution with the water movement is generally assumed. Being based on such experimental findings, the macropore domain in our model is defined based on this assumption, too. A deviation of solute and water movement during unsaturated transport is a common phenomenon. However by reducing the information to a binary existence of stained patches at a given depth level, the actual diffusive infiltration reach is not considered during the preprocessing.

*Is the methodology restricted to 1d vertical macropores? It seems to me that it could be extended to single macropores that have more complicated geometries.*

It is imaginable to extend the methodology to more complex geometries. However, this will require more information and explicit handling of the friction within such macropores. So far, different and more complex geometries are subsumed in a rather bold assumption based on the deviance of literature and theoretical values of the maximal transport velocity. Instead of handling complex geometries explicitly, we suggest to extend the data basis about such frictional losses in different structures and different states.

*The drift term (velocity) is not trivial to me. It is uniform inside a macropore when saturation is reached, it is not before saturation. How is the derivative of the diffusivity handled numerically with macro-pore – matrix interactions?*

The drift term is used to describe the directed, gravitationally driven component of the soil water movement in the matrix. It is not trivial in its implication on the processes, but rather straight forward with regard to its numerical application. Advection in macropores is describes as film flow constrained by the impulse balance with macropore matrix interaction. If the matrix is above field capacity, the frictional losses minimise and
particles start to be in concurrence for free slots for infiltration or sufficient freedom for exfiltration from and into the matrix respectively. Inside a macropore the film water is not considered for diffusive flow. This is justified by the subordinate time scales of diffusive processes during macropore advection. But the referee is correct that situations with saturated dead end macropores and slow infiltration might not be fully captured.

*p. 7, line 16: Where is the 0.7 percentile coming from?*

This percentile is hypothesised based on the assumption that all free bins need to be considered but do not contribute evenly. The number 0.7 was arbitrarily chosen to compensate for a skewed distribution here.

*Solutes are injected at high concentrations (5g/l KBr). Density effects may affect the fluid velocity.*

This is an interesting point. Currently, the model is foreseen to update the soil water viscosity based on temperature and pressure (which is held static at the moment). We agree that solute concentrations could also be considered. However, first test with dynamic viscosity resulted in rather subordinate effects as long temperature ranges near freezing were avoided. Actually, this is an additional capacity of the model that such hypotheses can be tested.

*I did not understood how the particle breakthrough is computed over the domain. Is it an arithmetic average ? Flux averaged ? This holds for water and contaminant BTCs.*

The Lagrangian approach allows us to flag and track any particle. For the breakthrough curves in the manuscript we calculated the depth-distribution of new particles of the respective event. As such it is representing the result of the described diffusive and advective soil water redistribution. With regard to contaminants we so far neglect particle interaction and diffusive dilution, which is assumed reasonable given the short time scales. Thus the water and solute breakthrough is calculated by converting the depth distribution of the number of new particles to water mass/volume or mass of based on

an initial concentration, respectively.

*The authors are slightly too enthusiastic by interpreting the simulation of the irrigation experiment. First, they provide a comparison for short time (infiltration over 20cm). At this time, it is difficult to identify biases. Second, there is a preferable flow which transported the tracer to a depth of 30-40 cm (see fig. 10) and which is not reproduced by the model. Despite these differences, I agree with the authors that their alternative model is able to simulate that experiment.*

We agree that the presented approaches are only a first step. Especially applications under different setups and further experimental references are needed and foreseen. Moreover, there are numerical concerns to be solved to enable longterm simulations. The model setup presented in Figure 10 could only be evaluated until 45 min after irrigation onset so far due to the high computational demand in the film flow routine. Thus the second hump in 30-40 cm did not yet establish in that time.

*Concerning the comparison with TDR measurements, how is the support volume defined and how is it taken into account in the modelling?*

The support volume of the IMKO IPH tube probe is very large. Especially the vertical extend of the signal guides of the probe (18 cm) is important here. The probe was manually applied and the mid point of the probe is taken as reference. It was lowered in increments of 10 cm. The measurement procedure was considered to calculate the references by averaging the total soil moisture of the depth increment referring to the respective probe depth.
* * *

---

## Author Comment (AC2) · 8 Feb 2018

We sincerely thank referee 2 for reviewing our manuscript and highlighting some important aspects, we should convey more clearly. This reply is a response in the sense of an interactive discussion. A response addressing the revision of the manuscript in detail will be given after the discussion phase.

*p. 6/7: The authors introduce here a relaxation time to LTE for a local non-equilibrium configuration of particles. What is the correspondence of this rule in a Eulerian frame, i.e., on the level of the Richards equation? Is it a type of first-order relaxation relation as used by Hassanizadeh and Gray for example?*

We agree that the work of Hassanizadeh and Gray is of high relevance to the field and

our approach. Gray and Hassanizadeh (1991) give one of the rare theoretical foundations of the energy states in unsaturated flow which go far beyond our simplification for LTE relaxation. As detailed in Hassanizadeh and Gray (1990, 1993) they developed a theory for multiphase flow in porous media combining averaging of microscale descriptions and macroscopic approaches by employing balance laws and the second law of thermodynamics. Although the general lines of thought are similar in our approach, we cannot claim to rigorously derive it from first principles alone. During the development of our model, we sincerely considered their raised concerns about interfacial exchange of momentum and their microscale description of matric potential as difference of the pressure of two fluids. However within the terminology of Hassanizadeh and Gray our macroscopic description scale might be even above their REV.

Our approach to conceptualize LTE relaxation time makes use of the energetic changes associated with momentum dissipation andÂăinfiltration of water from the macropore into the surrounding matrix to overcome the well known limitation of instant LTE in current Eulerian models. The corresponding rule to our concept in an Eulerian frame would be a temporal deviation from the state determined soil water retention curve in the case of infiltration. Through the use of water particles, we can achieve a representation of a faster fraction without artificially mobilising pre-event water bound in the soil capillaries as only the new particles experience this freedom. The particle approach in combination with the binned pore approximation also enables us to analyse the dynamics of LTE relaxation. The limit to analyse this relaxation better than our rough assumption appears to be still a lack of experimental references for the process as was also noted 25 years ago by Hassanizadeh and Gray (1993). Recently Schlüter et al. (2017) published very interesting new experimental insights at the microscale.

*Eq. (7): What is the mass of a water particle? The "water particles" are merely a conceptual picture, in fact, they correspond rather to "saturation particles". Equivalence between Richards equation in saturation form and the Langevin equation is achieved in the limit of infinite number of particles. The authors should clarify these points. Also*

*the assignation of a particle radius (how is this radius determined) to what conceptually is a point particle is unconventional.*

We agree that the use of particles is conceptual – similar to the use of particle tracking for simulating solute transport (these are not individual molecules). The equivalence between the the Richards equation and the Langevin equation is indeed when the number of particles approaches infinity (which physically does not make sense, as we have molecules).

We have shown the functional similarity between the Richards equation and the spatially explicit random walk of water particles (Zehe and Jackisch, 2016). In the model, the mass of a particle is defined by the setup of the model grid and the resolution of the porewater volume bins. This means, the finer the model grid and the better the required state resolution, the smaller the mass of water a particle represents. Obviously, this conceptual approach has limitations on both ends, when particles get too large or too small. Our test with different definitions so far did not result in massive deviations. However, we remained within "behavioural" bands and due conceptual test were left for further evaluation. Actually, we intend to reduce the number of required particles (or increase the representative particle mass) dramatically once the physical processes can be reduced to definitions of Markov-chains of higher order.

Referencing the particles as point masses with a volumetric footprint is indeed controversial. The matter arises from the combination of concepts, where the Lagrangian approach does not account for particle interaction with the solid phase but the Eulerian state control requires a translation into pore filling by means of a footprint. This receives an additional assumption in the calculation of infiltration from macropore films into the matrix, where the hypothetical radius of a spherical particle is assumed to be the threshold.

Because a particle can fill a certain fraction in the pore space which is referring to a certain capillary tension bin, I would not speak of "saturation particles" as such. The

concept is to implicitly resolve such pore-scale configurations within the soil matrix. In our first publication (Zehe and Jackisch, 2016) we especially detailed on this.

*Section 3.3.3 How do the rules established for the Macropore-Matrix interaction correspond to the dual porosity models by Gerke and van Genuchten, for example?*

We agree that due comparison to existing models is a valuable benchmark. Because the models require different data for their parameterisation, this task is not as trivial – especially because experimental references are scarce. We have done a series of tests to compare the model against an artificial "macropore" in a packed sand cylinder as presented by Germer and Braun (2015). These experiments are very similar to the simulations of Gerke and van Genuchten (1996).

They regarded exfiltration from an irrigated "macropore" into the surrounding matrix. For this a central vertical macropore (filled with coarse sand for stability) was installed in a half-cylinder filled with fine sand. The macropore was irrigated with constant flow rate until breakthrough at the bottom was reached. The exfiltration and diffusive redistribution was observed by means of time-lapse photographs and tensiometer monitoring. When parameterising echoRD according to the retention properties of the fine sand, we could reproduce the experimental observations (Fig. 1)

With respect to the rules for macropore-matrix exchange, we also can calculate a mean exfiltration time of a particle at the pore wall for different states of the surrounding soil matrix (Fig. 2).

*Sections 4.2 and 5.1: The authors refer here to generic application tests as benchmarks. It is not clear however against which benchmarks the model results are compared, or in other words, which are the benchmarks? I could imagine a 2D numerical solution of the Richards equation, for example.*

The benchmarks we refer to in the presented application tests are in accordance with the given aspects in section 4:

- capability to simulate 2D diffusive soil water redistribution of non-uniform states

- capability to simulate macropore-matrix exchange

- realistic sensitivity to antecedent state and soil physical parameters

- robustness of stochastic realisations of equal definitions of the representative macropore domain

- overall performance in reproducing observations of an irrigation experiment

We agree that comparisons to current model approaches have advantages with regard to the spectrum we could refer our approach to. However, we chose to emphasise experimental findings as benchmarks because the simulation of infiltration in structured soils is exactly the case, where the assumption of a well-mixed state and purely diffusive flow is critical.

References:

Gerke, H. H., & van Genuchten, M. T. (1996). Macroscopic representation of structural geometry for simulating water and solute movement in dual-porosity media. Advances in Water Resources, 19(6), 343–357. http://doi.org/10.1016/0309-1708(96)00012-7

Germer, K., & Braun, J. (2015). Macropore-matrix water flow interaction around a vertical macropore embedded in fine sand – laboratory investigations. Vadose Zone Journal, 14(7). http://doi.org/10.2136/vzj2014.03.0030

Hassanizadeh, S. M. & Gray, W. (1990). Mechanics and thermodynamics of multiphase flow in porous media including interphase boundaries. Advances in Water Resources 13, 169–186. http://doi.org/10.1016/0309-1708(90)90040-B

Hassanizadeh, S. M. & Gray, W. G. (1993) Toward an improved description of the physics of two-phase flow. Advances in Water Resources 16, 53–67. http://doi.org/10.1016/0309-1708(93)90029-F

Gray, W. G., & Hassanizadeh, S. M. (1991). Unsaturated Flow Theory Including Interfacial Phenomena. Water Resources Research, 27(8), 1855–1863. http://doi.org/10.1029/91WR01260

Schlüter, S., Berg, S., Li, T., Vogel, H.-J. & Wildenschild, D. (2017). Time scales of relaxation dynamics during transient conditions in two-phase flow. Water Resources Research 53, 4709–4724. http://doi.org/10.1002/2016WR019815

Zehe, E., & Jackisch, C. (2016). A Lagrangian model for soil water dynamics during rainfall-driven conditions. Hydrol. Earth Syst. Sci., 20(9), 3511–3526. http://doi.org/10.5194/hess-20-3511-2016

[Figure]

[Figure]

**Fig. 1.** Diffusive exfiltration from an irrigated artificial macropore. The irrigation rate was 3.78 L/h. Left panel: Model simulation of relative saturation (the half cylindrical column is assumed as planar

**exfiltration from macropore**

Fig. 2. Exfiltration time from macropore for different soils and matrix states.

---

## Author Response (AR1)

**Revisions on the manuscript by Jackisch and Zehe**
**Ecohydrological particle model based on representative domains**

Again, we express our gratitude to the constructive reviews and editing of our manuscript. We have carefully revised our manuscript throughout. With this reply, we explain the major lines of revisions.

Both reviewers highlighted deficits in the explanation of the conceptualisation of mobile water particles and the Lagrangian formalisms related to it. Moreover, it appears that the dual domain approach was not fully conveyed. To clarify this, we revised the respective descriptions in the introduction and methods section. Specifically, we extended the explanation of the particle concept and added more information about the resulting model behaviour in synthetic testcases in Appendix D.
Moreover, we detailed on the implicit conceptualisation of the soil water distribution in the pore space and the relaxation to local thermodynamic equilibrium after deviation during infiltration.

Following the suggestion of reviewer #1, we added explanation of the calculation of the breakthrough curves as depth distribution of new particles in the methods section 4.1. We also highlighted that we neglect diffusive mixing between the particles as a consequence.

Both reviewers demanded for more self-critical discussions of the results. We regret not to have conveyed our deep concerns about the limitations and drawbacks of the approach more clearly. We revised the discussion section in this regard clarifying the limitations and conceptual bounds.

All further suggestions of the reviewers have been addressed according to our responses before. However, we did not include a discussion about density effects in the particle approach, as this was not explicitly studied so far. Generally, the model can treat water with different viscosity and also changes in volume can be accounted for as the particles are referred to through a unit mass. But with dependence on the soil water retention curves for diffusivity and hydraulic conductivity and with missing particle interaction, we do not see that echoRD is ready for this.

We add a comparison pdf to the submission with highlighted changes below.

Sincerely,
Erwin Zehe and Conrad Jackisch

[revised manuscript text omitted]